# Protein Aggregation Landscape in Neurodegenerative Diseases: Clinical Relevance and Future Applications

**DOI:** 10.3390/ijms22116016

**Published:** 2021-06-02

**Authors:** Niccolò Candelise, Silvia Scaricamazza, Illari Salvatori, Alberto Ferri, Cristiana Valle, Valeria Manganelli, Tina Garofalo, Maurizio Sorice, Roberta Misasi

**Affiliations:** 1Fondazione Santa Lucia IRCCS, c/o CERC, 00143 Rome, Italy; silviascaricamazza@gmail.com (S.S.); illarisalvatori@libero.it (I.S.); alberto.ferri@cnr.it (A.F.); c.valle@hsantalucia.it (C.V.); 2Institute of Translational Pharmacology, National Research Council, 00133 Rome, Italy; 3Department of Experimental Medicine, University of Rome “La Sapienza”, 00161 Rome, Italy; valeria.manganelli@uniroma1.it (V.M.); tina.garofalo@uniroma1.it (T.G.); maurizio.sorice@uniroma1.it (M.S.); roberta.misasi@uniroma1.it (R.M.)

**Keywords:** intrinsic disorder, phase separation, protein aggregation, neurodegenerative disease, prion protein, alpha synuclein, TDP-43, tau, amyloid beta

## Abstract

Intrinsic disorder is a natural feature of polypeptide chains, resulting in the lack of a defined three-dimensional structure. Conformational changes in intrinsically disordered regions of a protein lead to unstable β-sheet enriched intermediates, which are stabilized by intermolecular interactions with other β-sheet enriched molecules, producing stable proteinaceous aggregates. Upon misfolding, several pathways may be undertaken depending on the composition of the amino acidic string and the surrounding environment, leading to different structures. Accumulating evidence is suggesting that the conformational state of a protein may initiate signalling pathways involved both in pathology and physiology. In this review, we will summarize the heterogeneity of structures that are produced from intrinsically disordered protein domains and highlight the routes that lead to the formation of physiological liquid droplets as well as pathogenic aggregates. The most common proteins found in aggregates in neurodegenerative diseases and their structural variability will be addressed. We will further evaluate the clinical relevance and future applications of the study of the structural heterogeneity of protein aggregates, which may aid the understanding of the phenotypic diversity observed in neurodegenerative disorders.

## 1. Introduction

The correct function of living organisms depends on the concerted effort of a network of thousands of proteins [1,2,3], which are required to assume a defined structure to exert their function. Protein folding is a tightly regulated process that relies on the primary sequence of the protein. The folding depends on the energetic landscape, as thermodynamic constrictions force a protein to assume the conformation that minimizes the free energy of the system, and on the activity of chaperones helping the protein to reach its functional structure. However, intrinsically disordered regions (IDRs), found in most eukaryotic proteins [4,5,6], appear to show no preference for chaperone binding [7]. Thus, proteins mostly or entirely composed of IDRs, defined as intrinsically disordered proteins (IDPs), may escape the classical folding pathway and maintain a metastable and plastic state. A subset of IDRs termed low complexity domains (LCDs), found in up to 1.2% of the protein-coding human genes [8], is gaining attention due to its presence in aggregated state in most neurodegenerative disorders. Mild conformational changes in LCDs may lead to an unstable β-sheet enriched intermediate, which is further stabilized by intermolecular interactions with other β-sheet enriched molecules, thereby producing the stable proteinaceous aggregate. This mechanism may be viewed as an unorthodox signalling pathway in which conformational information, rather than addition or remotion of functional groups, mediates the cascade of events leading both to physiological and pathological outcomes. Furthermore, neurons do not have the ability to remove or dilute toxic molecules by mitotic division as they are terminal cells; therefore, they are very sensitive to misfolded proteins, increasingly with aging.

In this review, we will first recapitulate how intrinsic disorder determines the formation of a variety of structures and produces different states of the matter, owning both physiological and pathological features. Next, we will provide evidence of structural variability observed in the proteins most commonly associated with neurodegeneration, specifically the prion protein (PrP), amyloid β-peptides (Aβ), tau, α-synuclein and TAR DNA binding protein 43 (TDP-43). As a caveat, we will focus on the wild-type sequence of these proteins, as most neurodegenerative diseases are sporadic in nature [9]. Besides, most mutations associated with neurodegenerative diseases map on the LCDs leading to an increased propensity to form misfolded species [9,10,11]. The clinical relevance and future applications of the study of the structural variability of protein aggregates will be addressed, as it may aid the understanding of the phenotypic diversity observed in human neurodegenerative disorders.

## 2. Intrinsic Disorder

Intrinsic disorder is a natural feature of polypeptide chains, resulting in the absence of a well-defined spatial organization [5,6,12]. The presence of IDRs appears to correlate with the complexity of the living organism [13,14], with half of eukaryotic proteins expected to possess at least one IDR in their sequence [4,5,15]. A plethora of biological functions are associated with IDRs [5,6,16], spanning from stress response [17] to high-order assembly [18] to RNA metabolism [19,20,21]. Broadly, IDRs may be classified in those who fold on binding or upon environmental changes and those that remain substantially unfolded [6,14,22]. A further level of variability resides in the ability of IDRs to assume non-native states of globular proteins such as the molten-globule or the coiled-coil state [5,6,23,24]. Being metastable and displaying weak and transient interactions [12,25], IDRs may change their structure in time, fluctuating among different low energetic levels.

Intrinsic disorder is encoded by the composition of the sequence rather than the exact string of amino acids [26,27]. Therefore, disordered structures may be achieved by a variety of sequences, which substantially differ from those of ordered, structured proteins [5,6,28,29]. Hydrophobicity, net charge resulting from the presence of a subset of specific disorder-prone amino acids (R, K, E, Q, S, P, G, A) [30] and low complexity [31] contribute in maintaining the natively unfolded structure of a protein, defining the IDR. Interacting modules of IDRs capable of undergoing disorder-to-order transitions have been defined as molecular recognition figures and short linear motifs [32,33,34,35]. Low mean hydrophobicity and high net charge have been reported to increase the disorder and reduce the solubility by reducing compaction and increasing electrostatic repulsion [27,36,37]. However, the presence of hydrophobic stretches was shown to increase the propensity of forming β-sheets [26,27,38,39], as in the case of the non-amyloid component found in the IDR of α-synuclein and in the low complexity C-terminal domain of TDP-43. Polar residues, such as glutamine and asparagine, are capable of forming assemblies through multiple weak side-by-side interactions [40,41], offering the driving force to form amyloids. This is the case of the polyQ stretch found in huntingtin protein in Huntington’s Disease and in ataxin protein in spinocerebellar ataxia [42,43]. Similarly, most physiological fungal IDRs show a high content in Q/N residues [12,44]. As extreme cases, homopolymeric chains such as polyglutamine, polyasparagine and polyglycine were shown to form amorphous assemblies and amyloid fibrils [45,46,47,48]. Lastly, polybasic stretches rich in arginine and lysine, displaying a high surface charge, may also produce assemblies, as in the case of the N-terminal domain of the cellular prion protein and the RGG domain of FUS [49]. Examples of strings that characterize IDRs and their properties are summarized in Table 1.

Therefore, it is apparent that the lack of structural arrangement may be achieved through different compositions and features, ultimately converging in the ability to coalesce into higher order assemblies. Overall, low complexity offers another angle to achieve the preservation of the unfolded state, increasing the probability of forming β-structures and allowing for the formation of homomeric and heteromeric assemblies. Moreover, polymerization into supramolecular structures may differ in the state of the matter, implying different routes of aggregation.

## 3. Phase Transition

The cellular milieu is an environment crowded with macromolecules, accounting for a total intracellular concentration of up to 400 mg/mL [50,51]. Within this crowded medium with very little available space, IDRs act as hubs for the interaction with multiple partners, being either other proteins and IDRs or nucleic acids [52,53]. Dynamic interactions between IDRs and other molecules, according to the polyvalent electrostatic model [6,18], allow for condensation and phase transition: once the proper conditions are met, the assembly of proteins (and eventually nucleic acids) demixes from the cytoplasm (or nucleoplasm) driven by non-covalent and transient interactions such as electrostatic, hydrophobic, hydrogen bonding, dipole−dipole, π−π, and cation−π interaction [28,54]. This process, referred to as liquid−liquid phase separation (LLPS), leads to the formation of distinct liquid droplets within a coexisting liquid milieu, forming membraneless organelles (MLOs) [55,56,57]. Such structures are characterized by the high concentration of their internal components (hence defining a highly reactive environment), rapid exchange with the surrounding milieu and marked anisotropy of IDRs [24,54]. The latter indicates that, whereas in a diluted phase (e.g., the cytosol) IDRs appear to be compacted, they assume an expanded state in the liquid droplet, maximizing the chances of interactions. Accumulating knowledge [5,6,12,58] is suggesting that the formation of MLOs is a rather common phenomenon in living organisms. The nucleolus [59,60], Cajal bodies [61], stress granules (SGs), RNA-protein granules and processing bodies [62,63] are just a few examples of physiological MLOs that could be found in living cells. In some cases, a further phase separation may be observed within a MLO, with demixing between two slightly different viscous compartments [24,64]. In most cases, the presence of IDRs appears to be the driving factor for the transient assembly of MLOs due to their flexibility and ability to form multivalent interactions [65,66]. Moreover, MLOs may quickly assemble and disassemble as a function of the fluctuation of the environment [8,16]. Post-translational modifications (PTMs), namely phosphorylation, ubiquitination, acetylation, glycosylation and SUMOylation, play an important role, as the alteration of the net charge or the addition of a steric component may influence the state of the matter of the condensate [67,68]. MLO formation thus seems to be a tightly regulated process, as suggested by the positive evolutionary selection of IDR motifs. Indeed, multicellular eukaryotes own a higher amount of IDRs than monocellular eukaryotes, eubacteria and archeas [5,15]. Persistent liquid droplets, however, may transition into a more viscoelastic condensate (as in the case of the nucleolus [69]) and eventually form gel-like structures no longer capable of exchanging components with the surrounding environment. Although IDRs in liquid droplets retain their unfolded state, the transition into a gel-like state increases the propensity of forming β-structures, paving the way for the transition into solid structures. Upon transition into gel structures and even within MLOs, LCDs may assume β-features [24,51,54,70]. Crystallization experiments [70] suggest that LCDs in hydrogels form kinked β-sheets, owning β-sheets running parallel to the length of the crystal, with kinks of glycine or aromatic residues (termed low-complexity aromatic-rich kinked segments) stacked in ladders. Kinks prevent the interdigitation of side chains across the β-sheet interface, thus maintaining the flexibility of the structure. Nucleation of oligomers may take place in these phase separated structures, leading to the emergence of aggregates. The formation of solids from liquid droplets is sometimes referred to as “aging” [37,66], which could underlie the pathogenic process associated with neurodegeneration.

## 4. The Paths towards Toxicity

The transition into a solid phase from a liquid droplet implies that a chaotic and disordered system (e.g., a liquid droplet) is organizing into ordered structures, reducing both the entropy of the system and the ability to exchange components with the environment. Whereas the formation and disassembly of MLOs is a dynamic and reversible process, solid aggregates are not able to revert back to the liquid phase and are often linked to cell death. Three hypotheses based on kinetic studies have been put forward to explain misfolding and aggregation. In the polymerization model [38,71], polymerization depends on slow and unfavorable interactions producing an oligomeric nucleus which rapidly assembles into larger polymers. Microtubule assembly, for instance, seems to follow this model [72]. The conformational hypothesis [2,73] assumes that the protein is stable both as folded and misfolded. Here, conformational changes are required to produce the amyloidogenic oligomers. Lastly, the conformation/oligomerization hypothesis [74] proposes an intermediate view, with conformational changes leading to the formation of an unstable amyloidogenic oligomer, which will eventually grow to produce the solid aggregate. Following the latter hypothesis, the misfolding of LCDs into β-strands produces the active monomer, which appears to be the initial stage in the paths towards toxicity. This initial misfolding step grants the ability to perform interactions with other β-strands belonging either to the same protein or to another monomer. β-strands form hydrogen bonds among their backbone, assembling into β-sheets. During the assembly of the β-strands into sheets, various oligomeric seeds are produced due to the topology of the interactions among strands (reviewed in [39]). β-sheets bond one another through interdigitation of residues, forming steric zippers either with the same (homosteric) or different (heterosteric) sheets [11,75], stabilizing the core of the assembly. Depending on the symmetry and orientation of the β-sheet, eight theoretical steric zippers have been proposed [76], further increasing the heterogeneity of structures. Although the most common structure found in natural amyloids is the parallel in-register β-sheet [75,76], other tertiary β-assemblies have been observed [9,77,78,79]. Starting from the same primary sequence, a protein may assume different β-enriched oligomeric conformations. These are often referred to as strains [44,73,80,81], a term borrowed from virology to indicate different structures produced by the same protein specimen. A growing body of evidence is indicating oligomers as the major toxic species during the amyloidogenesis that leads to neurodegeneration [82,83,84,85,86,87,88]. Indeed, the potential to effectively “seed” the aggregation, referring to the ability to contact and convert monomers and hence self-propagate, seems to be an emergent property of oligomers, defined as “prion-like behaviour” (detailed in the next section). Oligomers represent an additional source of variability, as they may either be on-pathway or off-pathway as a function of their capacity to grow by addition of monomers. On-pathway oligomers are defined by their competence to further incorporate monomers, whereas off-pathway oligomers are stable and lack the ability to seed the aggregation [89,90]. Self-propagation of the oligomers is achieved through the addition of monomers of the same type (homotypic), forming hydrogen bonds between β-sheets that strengthen the core [91,92]. Stacked β-sheets from different monomers produce the characteristic quaternary cross-β-conformation that strengthens the core of the structure. Nucleating oligomers further assemble into nascent protofilaments of 2−7 nm in diameter [93,94], which grow by incorporating monomers and associate one another by lateral interactions in various topologies. Thus, structures such as fibrils, ribbons and amorphous aggregates are observed both in different disorders and within the same pathology. In patients, the multiplicity of amyloid structures produced by the same protein may hence reflect different clinical frames and ongoing pathologies. An illustration of the stages of IDPs’ aggregation and the relative energetic landscape is depicted in Figure 1.

## 5. PrP: The Protein That Started It All

Historically, the concept of the self-propagation and infectivity of a protein was postulated for the scrapie agent, a factor causing neurodegeneration in sheep [95]. The scrapie pathology showed remarkably similar features both to Creutzfeldt–Jakob Disease (CJD) and Kuru. The former is a rare neurodegenerative disease while the latter was an endemic neurodegeneration affecting the Fore tribe in the highlands of Papua New Guinea linked to their cannibalistic rituals [96]. Both Kuru and CJD were shown to be transmissible to chimpanzees and hamsters [95,97]. Pioneering studies demonstrated that whereas procedures aimed to hydrolyze nucleic acids were unable to lower the transmission of the scrapie agent, denaturation of proteins could reduce the infectivity [95,98]. The term “prion” (proteinaceous infectious particle) was proposed for the scrapie pathology as sole causative agent of the disease. Biochemically, the prion derived from scrapie infected hamsters showed an unusual resistance to non-denaturing detergents and to proteinase K, resulting in 27–30 kDa bands after electrophoresis [95]. Improvements in the purification protocol and microscopy imaging [99,100] showed the presence of rod-like structures of various sizes and shapes with tinctorial properties (binding to Congo red dye [101]) similar to those of the amyloids found in neurodegenerative diseases. Thus, the now widely accepted “protein only” hypothesis was proposed for the self-replication of the PrP [95,102]: its propagation requires the formation of a homotypic complex between the two molecules [103], causing the conversion of the native cellular form (PrP-C) into the pathogenic conformation.

PrP-C is a glycosyl phosphoinositol anchored membrane glycoprotein. Its N-terminal moiety presents a variable number of octapeptide repeats enriched in Glycine residues [104,105,106] forming the LCD of PrP. Traditionally, low complexity regions displaying infectivity have been often referred to as prion-like domains [1,2,5,6,8,9,73]. For the sake of the coherence of the present work, in this review we will keep addressing these structures as LCDs. The C-terminal moiety of PrP-C forms a globular domain consisting of two α-helices and one two-stranded antiparallel β-sheet [107], linked to the N-terminal part by a middle hydrophobic core. Surprisingly, the amyloid region of PrP appears to be the folded C-terminal region (residues 90–231), which undergoes structural transition from α-helix enriched to the typical cross-β structure observed in amyloids [108]. In mice genetically lacking PrP (PRNP^0/0^, behaving and developing normally), the overexpression of a PrP depleted of the N-terminal octapeptide repeats restored susceptibility to infection [104]. However, it required a longer incubation time and resulted in a lower PrP titer compared to the overexpression of the full-length version. Insertion of multiple octapeptide repeats increases the flexibility of the structure, weakening the α-helix in the C-terminal domain, thus promoting aggregation [105,109]. This evidence suggests that, whilst the N-terminal LCD of PrP-C is not essential for prion replication, it does have an effect on the rate of prion accumulation. Outside of Kuru and CJD, other human pathologies termed transmissible spongiform encephalopathies have been linked to the misfolding of PrP [110], namely the variant CJD (the “mad cow” disease) [111], Gerstmann–Sträussler-Scheinker disease [112], fatal familial insomnia [113] and variably protease-sensitive prionopathy [114].

Strains of PrP were discovered in seminal studies conducted in goats intracerebrally infected with Scrapie sheep’s homogenates [115]. The “drowsy” strain and the “scratching” strain were determined based on differences in inoculation time and lesion pattern. Importantly, these differences were maintained through several passages. Although sometimes referred to as strains, the main clinical and neuropathological forms of PrP are defined by a polymorphism codon 129, either a methionine or a valine (thus producing MM, MV and VV) and by their mobility shift observed by Western blot following proteinase K digestion, producing a type 1 fragment of 21 kDa and a type 2 of 19 kDa [116]. Up to six combinations are hence theoretically possible among polymorphisms and types. However, MM1 and MV1 are phenotypically identical and thus grouped in a single class (MM/MV1), whereas MV2 and MM2 cases are further divided into subgroups according to the lesion pattern: predominantly cortical lesions (MV2C an MM2C), kuru-like plaques (MV2K), or both (MV2K+C), and thalamic lesions (MM2T) [117]. Cases of MM1 prion type showing white matter lesions were also reported [118]. Outside the genetic variance that characterizes the dimorphism at codon 129, both types of PrP are found together in more than one third of the sporadic CJD patients [119,120], indicating that multiple strains may coexist in the same patient. Moreover, type 1 was further shown by conformation stability immunoassay to segregate into two different strains termed T1^20^ and T1^21^ based on their proteinase K resistance profile and lesion pattern [121]. Off-pathway PrP oligomers [122] and nucleic acid-mediated LLPS [123] were characterized as well, further increasing the heterogeneity of possible PrP structures. Selective vulnerability of different neuronal populations may account for the formation of different strains within the same organism, as substantiated by the differences in lesion pattern. Overall, the classical studies conducted to delve into the peculiar properties of PrP paved the way for the central role of protein misfolding and transmission in neurodegenerative diseases.

## 6. α-Synuclein and Synucleinopathies

The involvement of α-synuclein in neurodegeneration was originally identified in the attempt to characterize neurofibrillary lesions in Alzheimer’s disease (AD) by the use of anti-tau antibodies, which labelled cytosolic proteins of approximately 19 kDa [124]. Its central role in Parkinson’s Disease (PD) was later revealed in genetic studies conducted on a family with early onset of PD [125] and further associated with Lewy bodies and neurites, the hallmark plaques found post-mortem in PD patients’ brains.

α-synuclein is a 140-amino acid IDP that has a low complexity, repeated N-terminal region, a hydrophobic central region termed non-amyloid-β component (NAC, a LCD as well) [126] and a negatively charged, disordered C-terminal moiety [125]. The N-terminal region presents seven imperfect repeats of eleven amino acids, possessing a conserved core of a consensus sequence KTKEGV [127]. Structural analyses showed that α-synuclein adopts through its N-terminal part an 11/3 helix lying along the surface of the membrane, half-buried in the phospholipid bilayer [128]. The role of the interaction of α-synuclein with the membrane is still unclear, as both physiological and pathological features have been reported [125,129]. The formation of oligomers with a high content of α-helices at the N-terminal moiety has been observed by Raman spectroscopy in the initial phase of α-synuclein aggregation [130]. This heterogeneous population of spheroid oligomers is supposed to be off-pathway, as oligomers may still be in equilibrium with their monomeric counterpart, thus not obliged intermediates of the amyloid form [131]. Nonetheless, when on-pathway, the content of α-helices is reduced upon formation of protofibrils in favor of β-sheet structures and cross-β interactions. Solid state nuclear magnetic resonance and cryoEM imaging showed that the amyloid core of α-synuclein oligomers presents a Greek key topology with in register hydrogen bonds along the fibril axis [132,133]. Furthermore, a recent work shows the ability of α-synuclein to perform LLPS in vitro [134]. Here, kinetic and biophysical examinations showed that, in presence of a crowding agent such as polyethileneglycole or metallic ions and liposomes, α-synuclein forms liquid droplets. Similarly, both PD-associated mutations and post-translational modifications (i.e., phosphorylation) caused the demixing of α-synuclein into droplets. These assemblies appear to consist mostly of unfolded monomers, although a small amount of both oligomers and fibrils was found within droplets as well. Mutagenesis experiments pointed to possible intramolecular interactions (electrostatic and hydrophobic) between the N-terminal and the NAC domain of α-synuclein as a major drive for the LLPS. In aging droplets transitioning into a gel-like state, the content of monomers is reduced in favor of an increase of fibrils, whereas the oligomeric portion remained mostly identical, suggesting the existence of a steady state of the oligomers. Structurally, transmission electron microscopy imaging and CD spectra indicated that oligomers are enriched in α-helices and, as expected, fibrils are enriched in β-sheets, indicating a dynamic equilibrium of various α-synuclein structures within the droplet [134]. Together, these results point toward a critical role for α-synuclein phase separation as nucleation stage for further aggregation.

Besides PD and AD (see below), the presence of α-synuclein amyloids has been detected in a subset of neurodegenerative diseases collectively called synucleinopathies, including PD-associated dementia (PDD), dementia with Lewy Bodies (DLB) and multiple systemic atrophy (MSA). Whereas the latter is well-defined and separated from the other diseases by the presence of α-synuclein aggregates in oligodendrocytes (termed glial cytoplasmic inclusions [135]), the distinction among PD, PDD and DLB is arbitrary, relying on the time difference of the onset of a series of clinical symptoms and are sometimes considered opposing edges of the same pathological spectrum [136]. However, outside the clinical frame, marked differences have been reported, such as the neurotransmitter systems involved and the presence of heterotypic aggregates [136], supporting the hypothesis that they are different diseases, probably reflecting a strain difference between the two pathologies. Both DLB and sporadic PD have been stratified in stages of progression of the pathology along connected regions with marked differences not only between PD and DLB, but even within the same pathology, defining a subset of DLB (brainstem, limbic and neocortical) by their route of spreading [137,138]. Indeed, the aggregation of α-synuclein into different conformations, together with differences in the vulnerability of neuronal populations toward α-synuclein aggregates, have been proposed as causative of the variety of clinical outcomes [139]. The presence of glial inclusions in MSA strongly suggests a strain typing of α-synuclein, as MSA strains show tropism toward glial cells outside of neurons [135]. Moreover, both the aggregation rate and the morphology of the final stage of the aggregate appear to differ, with PD and MSA forming twisted structures termed ribbons and DLB forming straight aggregates, termed fibrils [140], highlighting the striking differences among these similar diseases.

## 7. TDP-43

The heterologous nuclear ribonucleoprotein TDP-43 was originally discovered for its ability to bind the trans-active response element of the human immunodeficiency virus-1 [141]. Subsequently, it was detected in hyperphosphorylated and ubiquitinated state as the major component of proteinaceous deposits in post-mortem brains of patients affected by amyotrophic lateral sclerosis (ALS) and frontotemporal dementia (FTD) [142], as well as secondary pathology in other neurodegenerative diseases [143].

TDP-43 is a 414-amino acid protein containing two RNA recognition motifs, a nuclear localization and a nuclear export signal in its N-terminal moiety, and a glycine-rich (LCD) in its C-terminal moiety [144]. TDP-43 LCD presents two flanking IDRs enriched in hydrophobic (V, L, I, M) and aromatic (F, Y, W) residues. The LCD may adapt a helix-turn-helix conformation spanning residues 320–343, followed by two antiparallel β-sheets at residues 341–366 [144]. A physiological homo-oligomeric solenoid structure, in which the N-terminus and the C-terminus are physically separated, was proposed [145] as a structure that could halt the formation of pathogenic aggregates. Moreover, residues from 328 to 333 may form steric zippers characteristic of amyloids. Finally, the presence of aromatic residues in the IDRs allows the possibility of forming low-complexity aromatic-rich kinked segments, instrumental in the formation of MLOs [146]. Indeed, the ability of TDP-43 to perform LLPS and segregate into SGs is one of its best-characterized features [144]. TDP-43 is shuttled into the nucleus due to its nuclear localization signal, where it participates in a wide range of RNA metabolic steps [147]. Under stressful conditions, TDP-43 is exported from the nucleus to the cytoplasm, where it undergoes LLPS through its kinked helical C-terminal structure [146]. However, metabolic stress induced by lactate exposure revealed the ability of TDP-43 to coalesce into micronuclei in cultured cells [148]. Live cell imaging experiments showed that TDP-43 is able to perform nuclear LLPS in physiological conditions in a wide range of cellular models (neuronal-like cells, primary cultured mouse hippocampal neurons and human induced pluripotent stem cells). Moreover, under stressful conditions, researchers showed that TDP-43 forms LMOs in the cytoplasm both by associating to SGs and independently of SG markers, which may further evolve into gel-like and solid aggregates. Morphologically, SG-associated TDP-43 appears as a spherical droplet, whilst mature TDP-43 aggregates contain amorphous filamentous and skein-like structures [149]. Together, these results indicate that TDP-43 may follow different pathways of aggregation as a function of the environment and its subcellular localization [150,151].

In the clinical practice, the heterogeneity of symptoms and neuropathological lesion patterns observed in ALS and FTD lead to the classification of five putative strains, termed types A to E [152,153]. Type A is characterized by abundant dystrophic neurites and oval shaped neuronal inclusions in cortical layer II, typically observed in FTD associated with progranulin mutations; type B shows neuronal inclusions in all cortical layers and few dystrophic neurites; type C neuropathology presents long dystrophic neurites mostly found in cortical layer II of FTD cases; type D consists of widespread short dystrophic neurites and lenticular neuronal inclusions; lastly, type E is associated with ubiquitin-negative filamentous neuronal inclusions with wide neuroanatomical distribution, found in sporadic subset of FTD termed behavioral variant. Besides the neuropathological evidence, Western blot analyses conducted on patients’ brain homogenates revealed differences in the banding pattern of the C-terminal moiety of TDP-43 amongst various types, substantiating the putative strain typing of TDP-43. Moreover, morphologically distinct aggregates of TDP-43, reminiscent of those found in ALS, could be produced by adding extracts of ALS patients into HEK-293 cells [154].

Overall, TDP-43 displays a remarkable structural plasticity, being able to perform multiple distinct LLPS and different strains. Interestingly, TDP-43 inclusions were even found in muscle cells, a population being increasingly recognized as central in ALS etiopathogenesis [155,156]. This evidence indicates that TDP-43 inclusions have a tropism that goes beyond neurons.

## 8. Tau and Tauopathies

Tau is a microtubule associated protein involved in tubulin assembly and axonal transport [157]. Human full-length tau is an electric dipole, presenting an acidic N-terminal domain followed by a proline rich region and a basic C-terminal tail. The latter owns up to four imperfectly repeated strings of 31 residues (termed R1 through R4) separated by spacer regions involved in tubulin polymerization. Indeed, it appears that the most potent inducer of tubulin polymerization is the spacer region KVQIINKK between R1 and R2 [158]. The central proline-rich region serves for intermolecular interactions such as SH3-containing proteins and peptidyl-prolyl cis/trans isomerase, whereas the N-terminal moiety projects out of the microtubule bundle to interact with other cytoskeletal elements and membranes. This domain is composed of two repeats of 29 amino acids, termed N1 and N2 [158,159]. Alternative splicing gives rise to six tau isoforms, differing in the amount of N-terminal repeats (0N, 1N or 2N) and the inclusion of the C-terminal repeat R2 (producing 3R and 4R tau isoforms) [157,158]. The lack of the R2 sequence in 3R tau isoforms accounts for their weaker tubulin binding activity, as they lack the spacer KVQIINKK region [158,160]. The differential expression of these isoforms is a tightly regulated process both in time and space. For instance, the shortest tau isoform 0N3R is expressed during development, whereas bigger tau isoforms are mainly found in adult brains, with 1N variants being the most expressed [161]. Moreover, neurons with long axons and a large diameter such as peripheral projecting neurons express an additional N-terminal sequence producing an isoform referred as Big Tau [158,162].

Interestingly, different tau isoforms are associated with different forms of neurodegeneration, defining a subset of pathologies termed tauopathies. Hence, in Pick’s Disease only 3R tau is found, while in corticobasal degeneration and in progressive supranuclear palsy only 4R tau isoforms are present in degenerative brains [157,163]. In other tauopathies both forms could be found, although with different stoichiometry, as in the case of the behavioral variant of frontotemporal dementia (with 3R isoform overrepresented compared to the 4R), primary progressive aphasia and primary age-related tauopathy. Lastly, a further categorization can be made in order to distinguish between primary tauopathies (those aforementioned) and secondary tauopathies in which tau pathology is complementary to one or more other proteinaceous inclusions. This is the case of chronic traumatic encephalopathy and Alzheimer’s disease, both showing an equal ratio of 3R and 4R isoforms along with Aβ plaques.

Regardless of the domain included or excluded by alternative splicing, tau monomer is an IDP as shown both by computational and structural approaches [164,165]. Intramolecular interactions allow tau to assume a compact structure in solution, with the C-terminus folded over the microtubule binding domains and capped by the N-terminus moiety, described as a paperclip-like structure [166,167]. Transient β-structures can be formed in the R2 and R3 regions at the level of two hexapeptide motifs, termed PHF6* and PHF6 (VQIINK and VQIVYK, respectively) [168]. Anisotropy studies [169] revealed that tau monomers can reside in at least two different conformations, with the two hexapeptides either exposed or buried. However, even within these conformations, different tau strains can emerge, as observed in monomers derived from AD and corticobasal degeneration patients’ brains [170]. Similar to other IDPs, tau was shown to demix into liquid droplets, which may be on-pathway for solid transition into aggregates driven by different intermolecular interactions [171]. Decreasing salt concentrations promotes LLPS, suggesting a major role for electrostatic interactions [172]. However, truncation experiments showed that tau demixes in a salt-independent but hexadediol sensitive manner [173]. Since hexanediol resolves liquid droplets formed upon hydrophobic interactions, these experiments indicate that tau can perform a hydrophobic-driven LLPS, mediated by its C-terminal domain, in addition to the electrostatic-driven one. Heterotypic interactions with polyanionic compounds such as nucleic acids were also reported as a way to induce tau LLPS. Indeed, a recent work demonstrated the presence of tau in nuclear speckles in association with RNA [174]. Within droplets, double electron–electron resonance spectroscopy experiments revealed that tau assumes a more extended structure, similar to that observed in anisosomes formed by TDP-43 [175]. Although CD and NMR studies showed an increased propensity for forming β structures in liquid droplets, the cross-β content, as analyzed by thioflavin fluorescence, was lower than those found in tau amyloids, indicating that tau LLPS may not necessarily be on-pathway for the formation of aggregates [171].

PTMs of tau have been proposed as critical events for tau phase separation and aggregation [176]. Indeed, in post-mortem patients’ brains affected by tauopathies (regardless the type of pathology), tau appears to be hyperphosphorylated, ubiquitinated and acetylated. Neutralization of positive charges’ lysine residues (mostly in the N-terminal domain) by acetylation causes tau to detach from microtubule [177]. Nonetheless, hyper-acetylation dramatically reduces the ability of tau to form LLPS [178], probably due to the inability to form electrostatic interactions. Furthermore, tau could be acetylated in some of the same residues targeted by ubiquitination. Thus, a model has been proposed [163] in which acetylation competes with ubiquitination, in turn avoiding the proteasomal system and favoring tau accumulation. Phosphorylation of tau is the most abundant PTM both in physiological and pathological contexts, with more than 80 phosphorylatable residues identified to date [176]. Similar to acetylation, phosphorylation mediates the detachment of Tau from microtubules, while dephosphorylation promotes the interaction, describing a signalling mechanism that regulates cytoskeleton dynamics. Experiments conducted with hexanediol and by manipulating salt concentrations indicated that, similar to acetylation, phosphorylation of tau increases its propensity to form hydrophobic-driven LLPS and, conversely, inhibits electrostatic-driven LLPS [179]. Hyperphosphorylated tau has been observed as the major form of tau present in pathological aggregates [157,158,163,165,176]. Whereas some phospho-sites, such as Ser262, Ser356, and Ser396 [163,180], are found at basal levels and are likely to be key regulatory elements, others are exclusively detected in disease (e.g., Ser422, Ser396/Ser404 and Ser208 [163]).

The lingering presence of hyperphosphorylated tau inside droplets can cause the coalescence at the liquid interface, resulting in an increase of thioflavin fluorescence mirroring an increase in cross-β structures [179]. The first event in tau oligomerization is supposed to be a dimerization mediated by the association of two monomers in anti-parallel fashion, stabilized by intermolecular forces between the acidic N-terminal domain of a monomer and the basic C-terminal end of another monomer [181]. The core of tau assemblies was shown to be made of the aggregation-prone PHF6* and PHF6 regions [166]. Upon oligomerization, tau acquires the ability to act as a template for the conversion of its monomeric version, thus establishing a prion-like mechanism of spreading [182]. Surprisingly, this feature is achieved without a canonical low complexity sequence, suggesting a convergent selection of the aggregated structure starting from different amino acidic compositions. Recently, the structure of the main components of tau neurofibrillary tangles found in different tauopathies, namely paired helical filaments and straight filaments, has been resolved by cryo-EM [183]. In AD, these structures have a C-shaped morphology, whereas in chronic traumatic encephalopathy a hydrophobic pocket is found, producing an enlarged C-shape [184]. In Pick’s Disease, two types of filaments have been observed, owning an elongated morphology [185]. Lastly, corticobasal degeneration is characterized by a four-layered fold structure made up of 4R tau isoforms [186]. Importantly, the fibers derived from different individuals sharing the same pathology are indistinguishable one another, suggesting the existence of different strains that can differentially seed tau pathologies, resulting in variability in the neuronal structures affected, hence producing different clinical manifestations.

Overall, tdisplays high structural heterogeneity due to its alternative splicing, to the micro-environment in which aggregation takes place and to the tight regulation of PTMs. Surprisingly, the most common form of tauopathy and, in general, the most common form of dementia, Alzheimer’s disease, appears to actually be a secondary tauopathy, in which tau aggregation is downstream to other pathogenic events and coexist with aggregates made up of Aβ (a sine qua non conditions for AD diagnosis) and other misfolded proteins.

## 9. The Curious Case of Aβ

The Aβ peptides are small amino acidic chains derived from the proteolytic process of the larger amyloid precursor protein (APP), whose aggregation into plaques represents the major hallmark of Alzheimer’s disease along with tau-derived neurofibrillary tangles [186]. APP consists of a cytoplasmic C-terminus followed by a single transmembrane domain and an extracellular glycosylated N-terminus. Human APP can be processed through two different routes, defined as non-amyloidogenic and amyloidogenic pathways [187]. In the non-amyloidogenic processing, the enzyme α-secretase cleaves the extracellular domain, producing a short soluble N-terminal fragment and leaving a membrane-bound 83 kDa fragment [187,188,189]. This cleavage is regulated by electrical activity and upon muscarinic acetylcholine receptor activation, suggesting a role in synaptic activity [189]. The membrane-bound fragment is further processed by the γ-secretase complex, consisting of four different proteins termed presenilin 1 and 2, nicastrin and anterior pharynx defective-2 [187,190]. Presenilins, the active components of the γ-secretase complex, catalyze a proteolytic cleavage on aspartate residues within the hydrophobic 83 kDa domain, liberating a short fragment (3 kDa) and an intracellular domain. In the amyloidogenic pathway, APP is first cleaved by β-secretase, which produces a 99 kDa C-terminal moiety. This region is further processed heterogeneously by γ-secretase, resulting in the production of Aβ peptides of sizes ranging from 37 to 49 amino acids [187,191,192].

Although molecular dynamics simulation showed that Aβ peptides are IDPs [193], transient secondary structures have been reported for various fragments. Raman and infrared spectroscopy studies showed that peptides containing residues 1–28 fold in a polyproline-II α-helical structure that transition into β-sheet upon binding to phospholipid bilayers [194]. Likewise, Aβ fragments of various sizes were shown by NMR to display very different conformational states (see ref. [187] for an in-depth review of the structures Aβ fragments may adopt in solution). The two major forms of Aβ found in pathology are Aβ 1–40 and Aβ 1–42 [195]. Although these fragments only diverge for two C-terminal residues, their structure and their ability to grow into fibrils strikingly differs [196]. Indeed, NMR-guided simulations showed that Aβ 1–42 possess β-hairpins between residues 31–34 and 38–41, reducing the structural flexibility of the C-terminus [197]. On the other hand, in SDS micelle, Aβ 1–40 appears to be unstructured up to residue 14, followed by an α-helix between residues 15 and 36 with a kink at the level of residues 25–27. The transition from α-helix to coiled conformation is promoted by deprotonation of two acidic residues, in turn favoring the aggregation [198]. Overall, NMR structural studies [197] showed that Aβ peptides may populate multiple conformational levels ranging from α-helices to β-sheets, with rapid transitions among different structural features. Along with the C-terminal sequence GGVVIA, exclusive to Aβ 1–42 and longer fragments, the key residues for Aβ aggregation have been identified in the sequences KLVFFA at the N-terminal moiety, spanning residues 16–20 and GAIIGL (29–34) [75,193]. Remarkably, these three sequences are characterized by repeated residues with a hydrophobic nature.

3D structures of residues 15–42 within Aβ 1–42 reveal a cross-β sheet conformation arranged in a double horseshoe-like architecture. This structure, in which hydrophobic residues are maximally buried and residues 1–14 show β strand features, appears to be the most toxic and abundant in AD [187,199]. Molecular dynamics simulations combined with theoretical free energy calculations [193] suggested that the KLVFFA sequence is preferentially arranged in α-helical fashion in the monomeric form of Aβ, whereas it adopts β-sheets’ feature in the dimeric form. Here, α-helices are stabilized by intramolecular hydrogen bonds while intermolecular hydrogen bondings are predominant in the β sheet conformation. Beyond dimerization, the path of Aβ peptides toward aggregation involves a widely heterogeneous population of oligomers, spanning from low molecular weight assemblies such as dimers and tetramers to midrange aggregates, protofibrils and fibrils [200]. Variations in size and arrangement of Aβ 1–42 oligomers have been proposed as the main difference between late onset and rapid onset AD [201]. Moreover, solid state NMR experiments revealed Aβ off-pathway oligomers as well, with a distinct structure compared to on-pathway oligomers [202]. Characterization of oligomers through different experimental approaches showed high structural variability, from discoidal shapes devoid of β structures to antiparallel β-turn-β motif to collapsed coil [187,203]. A recent paper [196] followed the aggregation of synthetic Aβ 1–40 and 1–42 peptides by atomic force microscopy and time-resolved thioflavin fluorescence. A drastic difference in the lag-phase (the baseline of a sigmoidal curve, representing the time required for the reaction to take place) was observed, with Aβ 1–42 displaying a fast rate (approx. 0.5 h) of aggregation compared to the slower Aβ 1–40 (approx. 20 h). Micrographies performed by atomic force microscopy revealed the steps leading to the formation of fibrils. Aβ 1–42 appears as small spherical aggregates as baseline time point. At 0.5 h, bigger aggregates are prevalent whilst small spheres are observed at the edge of large assemblies, indicating a growth in size by lateral association. After 1 h of incubation, filament structures are observed, composed by fusion among big aggregates. These results suggest that the dominant phenomenon behind Aβ 1–42 aggregation resides in the fusion among small spheroid structures. On the other hand, Aβ 1–40 was observed over a longer time period, in agreement with the time resolved thioflavin assay. Although the fusion mechanism appears to be conserved in Aβ 1–40, a mosaic of different sizes is observed, with both big and small spheroid aggregates found at the end-stages of the reaction. When mature fibrils were assessed, Aβ 1–42 was shown to form thin and branched fibrils, whereas Aβ 1–40 produced thick and straight fibrils.

The extreme variability of Aβ structures at the low scale of monomers and oligomers mirrors the diverse landscape of high order aggregates and plaques found in human brains. Aβ deposits may be found either as diffuse, fibrillar, dense cored or cottonwool [204], as well as in association with cerebral blood vessels, the latter defining a pathology termed cerebral amyloid angiopathy [205]. Fibrillar and dense cored plaques often associate with dystrophic neurites and reactive astrocytes and microglia, forming a unity termed neuritic plaque [206]. These structures are hallmarks of AD and a mandatory presence for the definitive post-mortem diagnosis of the pathology [143]. Nonetheless, Aβ aggregates poorly correlate with the severity of the disease, better recapitulated by tau inclusions [207]. Furthermore, Aβ plaques are found in healthy subjects as a normal consequence of aging with a frequency comparable to those found in AD [208]. Indeed, Aβ aggregates alone appear to be insufficient to trigger neurodegeneration [143]. Still, the central role of Aβ in the onset of AD as early event for the downstream aggregation of tau is supported by overwhelming evidence based on genetic data acquired both from patients and experimental models. However, failures in therapeutic strategies focused on the amyloid cascade hypothesis have questioned this view [207]. Accumulated knowledge is pointing toward a multi-proteinopathy etiology of AD, as TDP-43 inclusions and α-synuclein aggregates are often found along with the classical Aβ and tau aggregates [143,209]. Recently, prion protein oligomers have been associated with rapid onset forms of AD [210]. Moreover, in primary age-related tauopathy, tau tangles follow the same spreading pattern of AD but fail to reach the neocortex in absence of Aβ plaques [211], indicating that Aβ pathology is necessary for the onset of more severe stages of the disease. Conversely, Aβ deposits have been well documented as co-morbidity in other prominent neurodegenerative diseases, such as DLB (85%) and PDD (55%) [143].

The simultaneous presence of Aβ deposits in various neurodegenerative diseases, along with the presence of Aβ plaques in healthy individuals and the lack of neurodegeneration associated with Aβ alone, may suggest a “chaperoning” role for Aβ fragments in assisting neurodegeneration, exacerbating the pathologic phenotype regardless of the main proteinaceous aggregating species. Overall, neurodegenerative diseases are multi-faceted pathologies in which IDPs interact with one another and with the environment in a narrow equilibrium between functionality and uncontrolled aggregation. Table 2 recapitulates the ability of the proteins discussed thus far to form higher molecular assemblies and summarizes the pathologies in which these forms could be found.

## 10. Protein Quality Control

As already mentioned [7], to prevent the accumulation of potentially pathogenic aggregates, neural cells make use of a series of chaperones capable of recognizing misfolded proteins by means of the exposed hydrophobic portions, thus guiding their correct folding. Proteostasis or homeostasis of proteins, through the protein quality control system (PQC), requires the prompt degradation and eventual recycling of aggregates and misfolded proteins. PQC includes several proteolytic systems, including ubiquitin-proteasome system (UPS), chaperone-mediated autophagy (CMA), and macroautophagy [212].

The UPS is the system responsible for the degradation of most of the misfolded proteins. These are conjugated with ubiquitin, then deubiquitinated, linearized and introduced into the proteasome, which degrades them into smaller peptides [213]. It is important to underline that the proteasome is particularly vulnerable to protein aggregates; in fact, the passage channel of this structure has a very small diameter (just over 10–12 angstroms) and this does not allow the digestion of aggregates that are difficult to linearize. Proteotoxicity resulting from decreased UPS activity could represent potential damage to neurons [214,215,216,217].

The degradation system by CMA is able to act on misfolded cytosolic protein without interfering with normal molecules. Target proteins of CMA include aggregates showing a specific degradation signal, the KFERQ sequence and substrates generated by post-translational modifications. These substrates may be entrusted to the CMA-mediated degradation system in lysosomes by interaction of the chaperone (mainly Hsp70 family) with the lysosome membrane molecule LAMP2A [218].

When aggregates show resistance to both the CMA and the UPS, autophagy comes into play. In the proteostasis of post mitotic neurons the role of autophagy is of fundamental importance. In fact, in these cells the cytotoxic proteins cannot be diluted by cell division and, consequently, a good quality control must be entrusted to specialized mechanisms. Efficacy of autophagy clearance has been shown to play an important role for neuronal homeostasis and maintenance. Thus, the restoration or promotion of autophagic function has been proposed as one possible approach to delay aging, including brain aging [219]. Moreover, several studies have shown that a number of signaling molecules responsible for regulating neuronal activity are localized in membrane lipid rafts [220], for example, neuroglobin, which is found in the lipid raft and is involved in neuronal survival mechanisms [221].

As a consequence, alterations in lipid rafts’ components have been hypothesized to contribute to the loss of neural function and potentially to the cell death/cell survival or autophagy balance associated with neurodegeneration. In particular, lipid rafts at mitochondria associated membrane (MAM) level are structures involved in a number of key metabolic functions, shown to be altered in neurodegenerations such as AD, PD and ALS [222,223]. Garofalo et al. demonstrated that MAM-associated lipid rafts could represent a physical and functional platform operating during the early steps of autophagic process. In fact, GD3, a core component of lipid raft-like microdomains, has been detected in immature autophagosomes associated with phosphatidylinositol 3-phosphate PI3P and LC3-II, as well as in autophagolysosomes associated with LAMP1 [224]. In addition, disruption of mitochondrial dynamics by the knocking down of strategic molecules associated to MAM’s lipid rafts including MFN2, GD3 or ERLIN1 significantly prevented autophagosome biogenesis and maturation [225]. In light of this evidence, a dysregulation during autophagosome maturation might drive the accumulation of protein aggregates and increase neurodegeneration (Figure 2).

It is possible to direct misfolded proteins prone to aggregation to the autophagic mechanism for lysosomal degradation thanks to the involvement of molecules that function as adapters, such as p62 and NBR1 [226]. P62, normally inactive, is activated by binding to other molecules, mostly at the level of the ER. After the accumulation of non-degradable autophagic cargoes, the chaperone molecules residing in the ER and participating in this signaling chain are arginylated and, via the N-terminal arginine residue, bind to the ZZ domain of p62 in the cytosol. Once bound, p62 undergoes a conformational modification that induces its polymerization and the interaction with LC3-II, a molecule anchored on the membrane of autophagosomes [218]. The autophagosome thus begins its load and, once completed, fuses with the lysosome to form the autolysosome, for the degradation by lysosomal hydrolases of both, load and p62.

The failure of the PQC system to remove misfolded proteins in the nervous system is the biochemical process behind most neurodegenerative diseases. During aging, deterioration in the PQC systems causes the failure of protein degradation, which may result in the accumulation of misfolded proteins. The successive modifications of structure and aggregation represent in many cases hallmarks of neurodegenerative diseases.

In light of this, it is clear that molecules involved in the clearance of misfolded proteins could represent new pharmacological targets, for example, by controlling the activation of CMA chaperones and adapters, as well as using autophagy inducers, which could be included in future therapeutic strategies for the improvement of neurodegenerative diseases.

## 11. Clinical Outlook and Concluding Remarks

Protein aggregation is an emerging concept in biology. Accumulating knowledge is suggesting that a convergent evolution has positively selected IDRs. Living organisms exploited the intrinsic property of IDRs to form amyloids to their advantage, incorporating it as a key signaling mechanism. Functional amyloids have indeed been documented throughout evolution, from yeasts to mammals [44,227]. Intriguingly, amyloid species of proteins associated with neurodegeneration may, in principle, serve a physiological role as well. For instance, proteinase K-resistant PrP was found in response to chronic morphine withdrawal in rats [228], whereas reversible, hyperphosphorylated tau tangles were found in brains of hibernating mammals [229]. The dysregulation of a physiological function associated with the aggregated state may thus represent the mechanism behind the abundance of IDR sequences found in proteins associated with neurodegenerative diseases (Table 3).

The understanding of the link between structural plasticity of proteins and neurodegeneration has led researchers to investigate this phenomenon as both a diagnostic and therapeutic tool. Particularly, the advent of dyes capable of specifically labelling aggregating structures [101,230], together with the exploitation of the prion-like seeded conversion mechanism, produced outstanding results in the early and differential diagnosis of neurodegenerative diseases. Protein misfolding cyclic amplification and real-time quaking-induced conversion assays have been applied to an increasing number of pathologies, starting with prion-related pathologies [231,232] and expanding to synucleinopathies [233,234], tauopathies and AD [235,236,237] and TDP-43-related pathologies [238], yielding a very a high diagnostic accuracy from ex vivo samples. Moreover, evidence of strain typing, as in the case of prion pathologies and synucleinopathies, has been documented with these technologies [239,240,241]. This group of evidence is particularly important in the clinical practice, as often neurodegenerative pathologies show overlapping phenotypes. For instance, atypical parkinsonism, characterized by motor symptoms derived from au misfolding and aggregation, have been accurately differentiated from synucleinopathies according to real-time quaking-induced conversion results [234]. Along with kinetic analyses, improvements in antibody-mediated detection of specific conformation are paving the way for the usage of a combination ratio of aggregation-related and neuronal damage proteins as diagnostic biomarkers [242,243,244]. On the other side, anti-amyloidogenic compounds are being studied as potential therapeutics [245,246,247]. Neurodegenerative diseases are still uncurable conditions. However, the basic knowledge of the biological role of the arrays of conformations may further aid the development of effective diagnostic and therapeutic tools by uncovering paths that could be relevant for the etiopathogenesis of neurodegenerative disorders.

High-throughput screening, combined with computer-based drug design, are employed for the discovery of novel therapeutic agents [248,249]. These methodologies are able to probe hundreds of thousands of drugs per day by scaling down the equipment and biological materials required to probe molecules. Typically, few hundreds of “hits” are selected for further studies on biocompatibility and availability, resulting in tens of molecules ultimately selected as “leads” for the development of a treatment. For instance, by applying biochemical high-throughput screening, combined with Thioflavin-T readouts, aminothienopyridazines were found to exert anti-amyloidogenic activity on tau aggregation by inhibition of fibril assembly [250].

However, high-throughput screening is still hindered by several factors. Most notably, cellular and biochemical platforms are simplified surrogates of the physiological environment that do not fully replicate the complexity of the organism. Efforts in the optimization of the system aimed to overcome this issue have led to the development of lab-on-a-chip and organ-on-a-chip technologies [251]. These newly developed exciting tools are miniaturized microfluidic perfusion devices in which primary cells can be grown for an extended time, allowing the screening of drug candidates in an environment similar to physiology, preserving cell-to-cell interactions, energy supply and removal of catabolites [248].

As we are just beginning to understand the phenomenon of phase separation and protein aggregation, the direction of the molecular pathogenetic study, an expression of basic science, seems right. This approach makes predictable a future in which these devastating diseases may be early and accurately diagnosed, so that personalized and disease-modifying therapies could slow down the insurgence of pathologies.

## Figures and Tables

**Figure 1 ijms-22-06016-f001:**
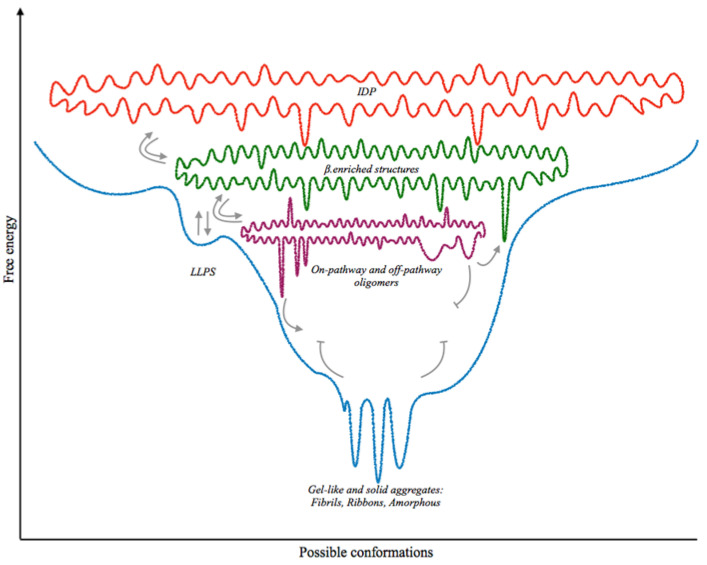
Illustration of the energetic landscape of IDPs. Model of energetic landscape of intrinsically disordered proteins plotted as a function of free energy. Unstructured IDPs (in red) fluctuate among multiple structural states with slightly different values of free energy. Structural transition to lower free energy states involves the formation of β-enriched structures (in green). The process is reversible (gray arrows) and may depend on overlapping energetic levels. Interaction among β-structures leads to oligomers (purple). LLPS may achieved at this stage (blue line depicting the overall trend of the aggregation pathway).These structures may be on pathway for the transition to stable, solid aggregates, or may be off-pathway (depicted as wider holes) which are more stable and cannot proceed further in the path towards aggregation, but may revert back to their monomeric form (indicated by the grey arrows). The final stage of aggregation (blue) consists of solid aggregates such as fibrils, ribbons and amorphous aggregates, which represent the absolute minimum of free energy a protein can own.

**Figure 2 ijms-22-06016-f002:**
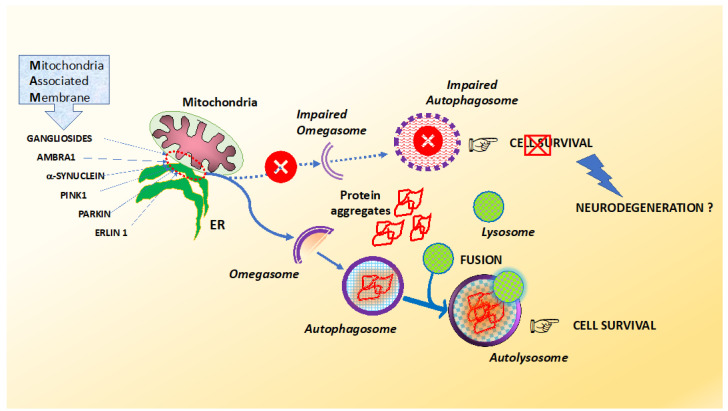
Efficient autophagy is required to remove protein aggregates and prevent neurodegeneration. Vesicle nucleation is regulated by molecular components associated to mitochondria-associated endoplasmic reticulum (ER) membranes (MAM), some of which are enriched in the lipid raft of MAM, including gangliosides, AMBRA1 and ERLIN1. Autophagosome formation is necessary to prevent the accumulation of damaged proteins and to ensure cell survival. The alteration of the autophagic machinery during the early stages prevents the maturation of omegasomes by hindering the autophagic flux and the clearance of aggregated proteins, thus contributing to neurodegeneration.

**Table 1 ijms-22-06016-t001:** Features and sequences of LCDs.

LCD Features	Type of Interaction	String Example	Protein Example	References
Hydrophobic-enriched (G, A, I, L, V, F, Y)	Hydrophobic	VTNVGGAVVTGVTAVA	α-synuclein	[26]
Poly-Q	weak side-by-side	≥8 Q residues	Huntingtin	[42]
Q/N-stretches	weak side-by-side, intra- and inter-molecular	P/QQGGYQQ/SYN repeats	[PSI^+^] (*S.cerevisiae*)	[44]
Homopolymers	H-bonds	GPG/GGX motifs	Flag (Silk protein)	[48]
Poly-basic	Net surface charge	RGRGG repeats	FUS	[49]

**Table 2 ijms-22-06016-t002:** Higher order structures and pathologies.

Protein	Strains	Off Pathway	LLPS	Associated Neurodegenerative Disease	References
Prion protein	Drowsy and scratching in sheeps; Type 1^20^, Type 1^21^, Type 2 in humans	V	V	Kuru, CJD, FFI, GSS, vPSPr	[111,112,113,114,116,121,122,123]
α-synuclein	MSA associated; PD/DLB associated; Fibrils and Ribbons	V	V	PD, DLB, MSA, AD *	[134,138,140]
TDP-43	Type A to Type E	/	V	ALS, FTD, AD *	[144,149,152,153]
Tau	CBD associated; AD associated	V	V	PiD, CBD, PSP, FTD, PART, AD	[157,170]
Aβ	Possibly associated to AD heterogeneity	V	/	AD, CAA, PDD *, DLB *	[201,204]

* co-pathology; / No evidence found in literature; FFI = Fatal Familial Insomnia; GSS = Gerstmann-Sträussler-Scheinker; vPSPr = variably Protease-Sensitive Prionopathy; PiD = Pick’s Disease; CBD = Corticobasal Degeneration; PSP = Progressive Supranuclear Palsy; PART = Primary Age-Related Tauopathy; CAA = Cerebral Amyloid Angiopathy.

**Table 3 ijms-22-06016-t003:** Amyloidogenic sequences.

Protein	Sequence(s) Features	LCD Sequence(s)	References
Prion protein	N-terminal octapeptide repeats	^51^PQGGTWGQ^58^	[105]
	C-terminal hydrophobic stretch	^112^MAGAAAAGAVVGGLGGYVLGSAM^134^	[106]
α-synuclein	N-terminal imperfectly repeated LCD	consensus sequence: KTKEGV	[127]
	C-terminal hydrophobic NAC domain	^61^EQVTNVGGAVVTGVTAVAQK TVEGAGSIAAATGFVKKDQL GKNEE^105^	[126]
TDP-43	Long C-terminal glycine-rich LCD	IDR1: ^216^RAFAFVTFADDQIAQSLCGEDLIIKGISVHISNAEPKHNSNRQLERSGRFGGNPGGFGNQGGFGNSRGGGAGLGNNQGSNMGGGM^310^	[144,145]
		Amyloidogenic core: ^311^NFGAFSINPAMMAAAQAALQSSWGMMGMLASQQNQSGPSGNNQNQGNMQ^360^	
		IDR2: ^361^REPNQAFGSGNNSYSGSNSGAAIGWGSASNAGSGSGFNGGFGSSMDSKSSGWGM^414^	
Tau	C-terminal hydrophobic PHF6 and PHF6* motifs	^275^VQIINK^280^; ^306^VQIVYK^311^	[166,168]
Aβ	N-terminal Hydrophobic-aromatic	^16^KLVFFA^21^	[75]
	C-terminal Hydrophobic	^29^GAIIGL^34^; ^29^GGVVIA^34^	

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
