# Peer review of "Protein Aggregation Landscape in Neurodegenerative Diseases: Clinical Relevance and Future Applications"

_ijms, 2021, doi:10.3390/ijms22116016_

Round 1

Reviewer 1 Report

Unfortunately, after a comprehensive review of this manuscript, I do not recommend it be accepted for publication. This manuscript requires extensive editing and restructuring of the English language and style. Throughout, some parts do not make sense, or the incorrect word has been used to describe what is being discussed. Also, throughout, there are statements made that are not supported by references and in many instances, outdated references have been used. I also have concerns that the authors have not correctly interpreted the literature and in parts provide very basic descriptions of key concepts that are not of the standard for publication. Several reviews on the same subject have been published in the last few years. The present manuscript does not seem to add any novel insight.

Author Response

R: We are truly sorry that Referee # 1 did not like the setting of our work. However, we took his comments seriously and tried to work on the manuscript following the suggestions indicated. In particular, we have restructured the final part to better focus the address of the work. We have revised the English and updated the References.

Reviewer 2 Report

In this paper titled” Protein aggregation landscape in neurodegenerative diseases: clinical relevance and future applications” the authors explained all the detailed structural features of different proteins like alpha-synuclein, TDP43, Tau, etc. which are responsible for the different neurodegenerative diseases, along with the mechanism of their aggregation. They also incorporated the importance of liquid-liquid phase separation (LLPS) in this mechanism. During this discussion, the low complexity domain (LCD) was discussed in order to understand their contribution towards these proteins’ aggregation propensity and entirely aggregation landscape. In the end, the story of protein quality control and clinical outlook complete all the aspects of the review. This review looks very interesting but there are few suggestions, which can improve this manuscript:

  • “………found in up to 1, 2% of the protein-coding human genes “in page 1 line 43, I am wondering whether it is 1-2% or 12 %?
  • On page 2 line 48, “This mechanism may be viewed as an unorthodox signaling pathway in which conformational information, rather than addition or remotion of functional groups, mediates the cascade of events leading both to physiological and pathological outcomes.” I saw that this sentence was repeated in the introduction and in the abstract.
  • I am wondering if the author could rephrase this sentence “three-dimensional structure within that sequence”.
  • “……once the concentration is high enough, the assembly of proteins “the concentration is not always required high, for multivalence interaction driven LLPS does not require high concentration. Sometimes, it requires only 1-10uM concentration.
  • More explanation and citation are required for this sentence “…….as suggested by the positive evolutionary selection of IDR motifs “on page 4 line 145.
  • Citation required “LCDs may assume ß-features” on page 4 line 151.
  • More explanation between LLPS and nucleation for aggregates would help the reader to understand the newly evolved concept of “The paths towards toxicity”.
  • In figure 1, the x-axis is missing.
  • I am wondering about the position of the LLPS in this landscape whether it will be a little bit upper than the on and off-pathway of oligomers. LLPS is sometimes reversible to monomer.
  • For all the proteins, disease mutants’ discussion is missing.

Author Response

  • “………found in up to 1, 2% of the protein-coding human genes “in page 1 line 43, I am wondering whether it is 1-2% or 12 %?

R: Sorry for the misprint! Thanks for pointing out: it's 1.2%, NOT 1,2%. We have corrected it in the text.

  • On page 2 line 48, “This mechanism may be viewed as an unorthodox signaling pathway in which conformational information, rather than addition or remotion of functional groups, mediates the cascade of events leading both to physiological and pathological outcomes.” I saw that this sentence was repeated in the introduction and in the abstract.

R: We thank the reviewer for pointing out the repetition. We accordingly changed the abstract section as follows:

“Accumulating evidence is suggesting that the conformational state of a protein may initiate signalling pathways involved both in pathology and physiology.”  

  • I am wondering if the author could rephrase this sentence “three-dimensional structure within that sequence”.

R: As pointed out, the sentence was rephrased as following in order to make it flow better. We thank the reviewer for the suggestion.

“Intrinsic disorder is a natural feature of polypeptide chains, resulting in the absence of a well defined spatial organization.”

  • “……once the concentration is high enough, the assembly of proteins “the concentration is not always required high, for multivalence interaction driven LLPS does not require high concentration. Sometimes, it requires only 1-10uM concentration.

R: We are grateful for raising this important point. Since the sentence is introductory, we chose to change the wrong sentence to a more generic one, as follows:

“once the proper conditions are met, the assembly of proteins”

  • More explanation and citation are required for this sentence “…….as suggested by the positive evolutionary selection of IDR motifs “on page 4 line 145

R: A further sentence and its reference has been added to better explain the concept:

MLOs formation thus seems to be a tightly regulated process, as suggested by the positive evolutionary selection of IDR motifs. Indeed, multicellular eukaryotes own a higher amount of IDRs than monocellular eukaryotes, eubacteria and archeas [5, 15].”

  • Citation required “LCDs may assume ß-features” on page 4 line 151.

R: We thank the reviewer for indicating the lack of important references, which have been added  ([24, 51, 54, 70]).

  • More explanation between LLPS and nucleation for aggregates would help the reader to understand the newly evolved concept of “The paths towards toxicity”.

R: We thank the reviewer for the suggestion. We added the sentence:

“Nucleation of oligomers may take place in these structures, leading to the formation of aggregates,” at the end of paragraph 3 before the “paths toward toxicity section.

We hope that the new sentece will help to better understand the concept.

  • In figure 1, the x-axis is missing.
  • I am wondering about the position of the LLPS in this landscape whether it will be a little bit upper than the on and off-pathway of oligomers. LLPS is sometimes reversible to monomer.

R: We thank the reviewer for the precious suggestion on the figure. x axis has been labelled and LLPS in the landscape has been moved upper, as suggested.

  • For all the proteins, disease mutants’ discussion is missing.

R: The discussion of the impact of mutations on proteins involved in neurodegenerative diseases was outside of the scope of the work. We reasoned that it would have required a significant amount of text to discuss them all, therefore we added a caveat at the end of the first paragraph (introduction), clearly stating that we would have not mentioned most of the mutations that occurred in neurodegenerative disease-relevant proteins.

Reviewer 3 Report

This manuscript submitted as a review by Candelise et al highlight the importance of several intrinsic disorder proteins found in humans and its pathological aggregation that leads to hallmarks of several neurodegenerative diseases. The overview presented here is thorough and includes appropriate information which will be of high interest to the readership of the International Journal of Molecular Sciences. The authors also have done a good job at citing relevant references. I only have minor comments and the publication of this manuscript is recommended after a minor revision. The authors are invited to address the following comments:

1) Page 9, Line 412, the sentence “up to four imperfectly repeated strings of 18 residues…” is incorrect. Tau protein repeats R1, R2, R3, and R4 each is about 31 residues long. Replace “18” with “31” in this sentence.

2) Page 17, Table 3, many of the sequence numbering are incorrect. For e.g. in α-synuclein, the C-terminal hydrophobic NAC domain sequence starts with 61 and ends with 95 instead of 105.  Similarly, the Tau sequence started with 276 and ended with 290 instead of 275, and ending with 280. Double-check for same in TDP-43 and in Aβ and correct these errors.    

Author Response

1) Page 9, Line 412, the sentence “up to four imperfectly repeated strings of 18 residues…” is incorrect. Tau protein repeats R1, R2, R3, and R4 each is about 31 residues long. Replace “18” with “31” in this sentence.

R: We thank the reviewer for pointing out the mistake. It has been fixed to 31.

2) Page 17, Table 3, many of the sequence numbering are incorrect. For e.g. in α-synuclein, the C-terminal hydrophobic NAC domain sequence starts with 61 and ends with 95 instead of 105.  Similarly, the Tau sequence started with 276 and ended with 290 instead of 275, and ending with 280. Double-check for same in TDP-43 and in Aβ and correct these errors.    

R: We thank the reviewer for highlight the mistake. After a thorough check, changes have been made to the table as follows:

Change reference on PrP: 104 -> 106;

Change on synuclein NAC residue number: 95 -> 105

Change on Tau PHF6 residue number: 276 - 290 -> 275 - 280

Add reference on Tau  PHF: [166]

On Abeta C-terminal: substitute the second sequence with  29GGVVIA34

Reviewer 4 Report

Dear Editor,

The manuscript by Candelise et al. reviews the heterogeneity of structures that are produced from intrinsically disordered protein domains and highlight the routes that lead to the formation of physiological liquid droplets as well as pathogenic aggregates in neurodegenerative diseases.

The review is comprehensive, informative and up-to-date (in most parts). Authors were successful in providing some well compiled opinions and summaries. The mechanistic figures and the addition of some suggestions for future directions in the conclusion section can be a good starting point for future studies and will be of interest for IJMS readers and beyond. I can endorse this manuscript for publication following the successful addressing of the minor edit below.

All the best!

Minor:

-Neurodegenerative diseases are yet incurable conditions. The author needs to briefly discuss future directions towards the end of their discussion and conclusion. This could include, but not limit to, use of high-throughput screening and computer-aided drug design as have been nicely reviewed by Aldewachi et al 2021 and Salman et al 2021 as they can provide a novel insight that can support the target validation of kinases in future studies. References to be included:

https://pubmed.ncbi.nlm.nih.gov/33672148/

https://pubmed.ncbi.nlm.nih.gov/33925236/

Author Response

Neurodegenerative diseases are yet incurable conditions. The author needs to briefly discuss future directions towards the end of their discussion and conclusion. This could include, but not limit to, use of high-throughput screening and computer-aided drug design as have been nicely reviewed by Aldewachi et al 2021 and Salman et al 2021 as they can provide a novel insight that can support the target validation of kinases in future studies. References to be included:

https://pubmed.ncbi.nlm.nih.gov/33672148/

https://pubmed.ncbi.nlm.nih.gov/33925236/

R: We thank the reviewer for the suggestion that will certainly improve the quality and completeness of our manuscript. We added sentences to briefly point to high-throughput screening and computer assisted drug discovery, as follows:

(…) On the other side, anti-amyloidogenic compounds are being studied as potential therapeutics [245-247]. Neurodegenerative diseases are still uncurable conditions. However, the basic knowledge of the biological role of the arrays of conformations may further aid the development of effective diagnostic and therapeutic tools by uncovering paths that could be relevant for the etiopathogenesis of neurodegenerative disorders. High-throughput screening, combined with computer-based drug design, are employed for the discovery of novel therapeutic agents [248, 249]. These methodologies are able to probe hundreds of thousands of drugs per day by scaling down the equipment and biological materials required to probe molecules. Typically, few hundreds of “hits” are selected for further studies on biocompatibility and availability, resulting in tens of molecules ultimately selected as “leads” for the development of a treatment.

For instance, by applying biochemical high-throughput screening, combined with Thioflavin-T readouts, aminothienopyridazines were found to exert anti-amyloidogenic activity on tau aggregation by inhibition of fibril assembly [250]. However, high-throughput screening is still hindered by several factors. Most notably, cellular and biochemical platforms are simplified surrogates of the physiological environment that do not fidely replicate the complexity of the organism. Efforts in the optimization of the system aimed to overcome this issue have led to the development of lab-on-a-chip and organ-on-a-chip technologies [251]. These newly developed exciting tools are miniaturized microfluidic perfusion devices in which primary cells can be grown for an extended time, allowing the screening of drug candidates in an environment similar to physiology, preserving cell-to-cell interactions, energy supply and removal of catabolites [248].

As we are just beginning to understand the phenomenon of phase separation and protein aggregation, the direction of the molecular pathogenetic study, an expression of basic science, seems right. This approach makes predictable a future in which these devastating diseases may be early and accurately diagnosed, so that personalized and disease-modifying therapies could slow down the insurgence of pathologies.